# Interventional, Quasi-Experimental Study of a Chronic Obstructive Pulmonary Disease Education Care Plan for Hospital Discharge

**DOI:** 10.3390/pharmacy9040202

**Published:** 2021-12-16

**Authors:** Letitia N. Warunek, Nicole E. Cieri-Hutcherson, Brian P. Kersten, Amany K. Hassan

**Affiliations:** 1Nesbitt School of Pharmacy, Wilkes University, Wilkes-Barre, PA 18766, USA; letitia.warunek@wilkes.edu; 2School of Pharmacy and Pharmaceutical Sciences, University at Buffalo, Buffalo, NY 14214, USA; 3Department of Pharmacy, Buffalo General Medical Center, Buffalo, NY 14293, USA; bkersten@kaleidahealth.org; 4School of Pharmacy, D’Youville, Buffalo, NY 14201, USA; hassana@dyc.edu

**Keywords:** COPD, hospital discharge, inhaler education, medication optimization, transitions of care

## Abstract

Chronic obstructive pulmonary disease (COPD) is one of the leading causes of morbidity, mortality, and reduced quality of life for patients. Proper use of inhaler devices is critical for effective drug delivery and prevention of COPD progression. The primary endpoint of this study was a mean percent increase in correct steps associated with inhaler technique after pharmacist education. The co-primary endpoint was a 25% increase in the proportion of patients correctly identifying the appropriate use of short-acting versus long-acting inhaler types. This was an interventional quasi-experimental study of patients hospitalized at a 491-bed tertiary academic medical center with a COPD exacerbation to assess a pharmacist-led COPD care plan. Eligible patients included general floor, adult patients admitted with a primary diagnosis of COPD exacerbation. The primary investigator recorded initial inhaler technique scores through a paper checklist, and provided education about device types and usage. Patients were reassessed within 48 h to determine if pharmacist education improved inhaler knowledge. A total of 67 patients received the COPD care plan before hospital discharge. At baseline, patients scored a median of 81.8% (67.5–97.0) of steps correct across all inhaler device types. After pharmacist education, patient scores increased to a median of 100% (90.9–100.0) (*p* < 0.0001). The proportion of patients correctly identifying when to use short-acting versus long-acting inhalers increased from 73.1% to 98.5% (*p* < 0.0001). Implementation of a pharmacist-led care plan for patients admitted for COPD exacerbation was associated with an increase in correct steps for appropriate inhaler technique and understanding of inhaler device types after pharmacist education.

## 1. Introduction

Chronic obstructive pulmonary disease (COPD) is a chronic disease, characterized by respiratory symptoms and airflow limitations. It is often the result of exposure to noxious particles or gases [1]. COPD is one of the leading causes of morbidity, mortality, and reduced quality of life for patients [1,2]. COPD accounts for approximately 800,000 hospitalizations annually, leading to nearly 50 billion dollars in health care spending [3]. Notably, up to 20% of patients hospitalized for COPD exacerbation are subsequently re-hospitalized within 30 days post-discharge, resulting in the third most costly health care expenditure for Medicare beneficiaries [4].

In response to increased hospital readmissions in the United States, the Centers for Medicare and Medicaid Services (CMS) introduced an initiative in 2011 to transition from fee-for-service payments to bundled payments for multiple services to create incentive for providers to provide high-quality care while reducing health care costs [5]. The focus of this transition was to reduce hospital readmissions within 30 days of hospital discharge for specific patient populations; COPD was added as a target population in October 2014 [6,7]. In order to meet the requirements for CMS bundled payments, hospitals must utilize evidence-based strategies to reduce hospital readmissions for target patient populations. Multiple strategies have demonstrated benefits in reducing hospital readmissions for other disease states such as heart failure; however, evidence for use in COPD remains limited [8,9].

Multiple review articles describe the emerging role of the pharmacist in the selection of inhalers and leading initiatives to implement patient education on proper inhaler use [10,11,12]. Proper use of inhaler devices is critical for effective drug delivery and prevention of disease progression and is emphasized as a key method to improve COPD management [1]. Inconsistencies and errors in the use of inhaler devices have been observed in various studies [13,14,15]. One prospective, cross-sectional study evaluated patient administration technique with common inhaler devices, including the pressurized metered dose inhaler (pMDI), the pMDI with a volumatic spacer, the Accuhaler^®^, and the Handihaler^®^. Results revealed that approximately 75% of patients performed at least one step incorrectly prior to education. Researchers concluded that formal face-to-face inhalation technique training led to a statistically significant reduction in percentage of incorrect techniques for all inhaler devices [16]. In addition, there is growing literature that describes a positive correlation between poor inhaler administration and increased health care utilization [17].

A pharmacist-led care plan focused on providing accurate inhaler education for administration was implemented at a 491-bed tertiary academic medical center. The goal of this care plan was to reduce the number of errors associated with inhaler technique for patients admitted with COPD exacerbations and to align with other ongoing COPD hospital readmission initiatives. The purpose of this study was to evaluate the impact of this pharmacist-led care plan on inhaler technique for patients admitted with COPD exacerbation.

## 2. Materials and Methods

### 2.1. Study Design

This was an interventional quasi-experimental study of patients hospitalized with COPD exacerbation to assess a newly implemented pharmacist-led COPD care plan at the study institution. This study was a one-group, pre-test/post-test design that was non-randomized. Implementation of a pharmacist-led COPD care plan focused on accurate inhaler device education was implemented in December 2018 at a 491-bed tertiary academic medical center. This care plan targeted patients admitted with COPD exacerbation who were diagnosed with COPD prior to hospital admission. The educational approach was implemented to align with other hospital initiatives focused on COPD. Exemption status for this study was granted by the local Institutional Review Board.

### 2.2. Participants

Patients carrying a diagnosis of COPD were included if they were admitted during a three-month period from December 2018 to February 2019 with an acute COPD exacerbation while residing on the general hospital floors prior to discharge. Patients admitted to the intensive care unit were screened for inclusion once transferred to the general floors. Daily screening occurred for active orders for intravenous methylprednisolone or oral prednisone (greater than or equal to 40 mg). An indication of COPD exacerbation documented in the electronic medical record associated with the corticosteroid orders was required.

Patients were excluded if they: had a past medical history of active lung cancer or bronchiectasis; had the presence of an artificial airway; or, had no true diagnosis of COPD. As additional adapted materials were not readily available, patients were excluded from receiving the care plan if they had certain physical limitations preventing teaching as per the research protocol (English not the primary language, deaf, blind). Patients were also excluded if they had significant dementia, resided in a nursing home or skilled living facility, and if patients were transferred to another facility prior to discharge.

### 2.3. Intervention

A summary of the study design and pharmacist-led COPD intervention is included for review (Table 1). The Joint Commission of Pharmacy Practitioners recognizes a pharmacists’ patient care process as a patient-centered approach to collaborate with other health care providers in order to optimized medication outcomes. As part of this process, pharmacists may develop an individualized patient-centered care plan, implement the care plan, and follow-up on this care plan to provide comprehensive care as part of a team [18]. The novel care plan used in this study was created by a pharmacist within the health system and was reviewed by the Pharmacy and Therapeutics Committee at the institution. The care plan was piloted on less than 10 patients in order to capture all device types during the pilot phase.

As no report or marker is available at our institution for identification of COPD exacerbation in real time, a report of high-dose steroid use was generated as a surrogate marker for patients who may be admitted for COPD exacerbation. Patients were identified via active orders for intravenous methylprednisolone or prednisone 40 mg or more with an indication for COPD exacerbation. Once patients were identified, the patient’s severity of COPD was classified using The Refined ABCD Assessment Tool within the 2019 Global Initiative for Chronic Obstructive Lung Disease (GOLD) Guideline recommendations [1]. A history of exacerbations, home oxygen requirements, and home inhaler medications were obtained for each patient to assist in classification. Once the patient was classified based on GOLD Guidelines, the patient’s inhaler regimen was assessed for appropriateness based on disease severity. Discrepancies in therapy and recommendations for therapy adjustment were discussed with the primary team prior to pharmacist-led education and hospital discharge.

If patients were identified as meeting inclusion criteria but were discharged prior to intervention, they were not included in the analysis. For remaining included patients, prior to hospital discharge, the pharmacist confirmed the inhaler regimen with the primary provider and then provided individualized inhaler education to the patient using demonstration inhaler devices. Inhaler device assessment was performed if the patient had used the device type in the past in order to determine the patient’s baseline understanding of inhaler technique prior to hospitalization. Scored checklists were prepared for each device type, and were adapted from package inserts and other available inhaler checklists [19]. The checklists had between nine and 14 steps to be assessed based on each inhaler device, and allowed for assessment of patient understanding of when to use short-acting versus long-acting inhalers (see Appendix A). The following device types were assessed: pressurized metered dose inhaler (pMDI), Respimat^®^, HandiHaler^®^, Ellipta^®^, Diskus^®^, and Respiclick^®^ (Appendix B).

The primary investigator delivered all components of the COPD care plan in order to reduce heterogeneity in data collection and patient counseling. A baseline knowledge assessment of each inhaler type was obtained using the scored checklist. Face-to-face training on inhaler administration was conducted at the patient bedside and educational handouts for each inhaler device were provided by the pharmacist. A follow-up assessment using the same scored checklist was performed within 48 h of the initial face-to-face training session to evaluate the effectiveness of the first encounter. Additional education was provided after the second patient interaction if required.

Additional pharmacist interventions to promote optimization of COPD management including adjustment of inhaler therapies, requesting inhaler refills at the time of discharge, and recommendations to address inhaler cost issues were recorded to evaluate additional benefits of the care plan. The pharmacist’s time spent delivering each component of the care plan was documented in order to quantify the impact of integrating this service into the daily responsibilities of the pharmacist.

Patient demographics including age, sex, presence of comorbid conditions, and smoking history were collected. COPD-related factors collected included number of previous exacerbations within the previous year, oxygen requirement prior to admission, and inhaler medications prior to admission. The patient’s understanding of inhaler administration was assessed using the same scored checklist for the baseline and follow-up assessment, and was obtained by the primary investigator in order to avoid inter-rater variability.

### 2.4. Outcomes

The primary endpoint of this study was a mean percent increase in correct steps associated with inhaler technique after pharmacist education. As described in a recent systematic review by Mahon et al., previous studies demonstrating the impact of pharmacist-led inhaler education lack standardization among study methods, including checklists, inhaler device types, and characterization of incorrect inhaler use [20]. The co-primary endpoint of this study was a 25% increase in the proportion of patients correctly identifying the appropriate use of short-acting versus long-acting inhaler types. A 25% increase was chosen in order to account for a higher baseline understanding of the inhaler types, as patients were required to have used the device type prior to hospital admission. Secondary endpoints included the frequency of errors associated with each type of inhaler device and number of additional pharmacist interventions to follow guideline-directed therapy. In addition, time spent on specific elements in the care plan was documented in order to evaluate the pharmacist time required to implement the care plan.

### 2.5. Data Collection

Paper copies of the checklist were used during patient intervention at the bedside. Data collected on the paper copies were transferred to an excel document after complete delivery of the care plan to the patient. These data were codified. Codified data will be kept indefinitely. Patient reports containing identifiable patient information that were created for this study were destroyed at study completion approximately 6–9 months after study approval.

In addition to information included on the checklists (Appendix A), patient demographics and clinical characteristics were collected on patients. The number of pharmacist interventions and the type of intervention were also collected for further evaluation.

### 2.6. Statistical Methods

In order to detect a 25% increase in the proportion of patients identifying the correct use of short-acting versus long-acting inhaler use after pharmacist education, a sample size of 35 patients was required for a power level of 90% and two-tailed test at a significance level of 0.05. The primary endpoints were analyzed using the Wilcoxon Signed-Rank test for continuous data and McNemar’s test for categorical data [21,22]. Descriptive statistics were used to summarize patient demographics.

## 3. Results

### 3.1. Participants

There were a total of 155 patients with an active order for intravenous methylprednisolone or oral prednisone greater than or equal to 40 mg residing on the general hospital floors during the study period. The majority of patients excluded did not carry a diagnosis of COPD and were on high-dose steroids for another indication or were not using inhalers prior to hospital admission (Figure 1). Of the 85 patients who met all inclusion criteria, 18 patients were discharged from the hospital prior to receiving the intervention. A total of 67 patients received all parts of the care plan, with a total of 136 individual inhaler device assessments. Prior to admission, 42 patients had been receiving triple therapy with a long-acting beta_2_-agonist, long-acting muscarinic antagonist, and an inhaled corticosteroid (Table 2).

### 3.2. Primary Outcomes

At baseline, patients scored a median of 81.8% (67.5–97.0) of steps correctly across all inhaler device types (Table 3). After pharmacist education, patient scores increased to a median of 100% (90.9–100.0) (*p* < 0.0001). The proportion of patients correctly identifying when to use short-acting versus long-acting inhalers also increased from 73.1% to 98.5% (*p* < 0.0001).

There was a notable increase in correct steps across all inhaler device types (Figure 2). The pMDI was the most common device type evaluated, with the most common error being failure to hold the device upright and shake well prior to use (Table 4). There was a statistically significant improvement in the three most common errors associated with the pMDI and Respimat^®^ devices, and for the most common error associated with the HandiHaler^®^ device.

### 3.3. Secondary Outcomes

In addition to providing inhaler education as described in the care plan, additional pharmacist interventions were identified for 36% (n = 25) of the patients (Table 5). The most common interventions included optimizing inhaler medications according to the patient’s GOLD classification and notifying providers of needed refills prior to hospital discharge. During education sessions, the pharmacist was able to provide education for smoking cessation for six patients. The greatest amount of time spent implementing the care plan was completing the baseline assessment of inhaler administration and providing the initial inhaler education (Table 6). The average time spent per patient was 44.3 ± 6.4 min.

## 4. Discussion

### 4.1. Primary Outcomes

This interventional quasi- experimental study evaluated the impact on improvement in inhaler technique for patients hospitalized with COPD exacerbation who received a pharmacist-led COPD care plan focused on providing education on correct inhaler administration. Implementation of this pharmacist-led COPD care plan was associated with an 18% increase in correct steps in inhaler technique after pharmacist education. In addition, there was a 25% increase in the proportion of patients who were able to identify the correct use of short-acting versus long-acting inhaler types.

This study included patients admitted for COPD exacerbation who had a diagnosis of COPD and previous use of inhalers prior to hospital admission. These criteria were included in order to identify opportunities for intervention in patients already being managed for COPD in the outpatient setting. Exclusion criteria were designed to avoid including patients that could have other reasons for pulmonary decompensation requiring hospitalization. Implementation of this care plan demonstrates a continued need for inhaler education and reassessment for patients after initial diagnosis of COPD, which is consistent with previous studies. A study conducted by Ahn et al. evaluated the benefits of repeat inhaler education and quality of life in patients with previously diagnosed COPD over a six month time frame [14]. The intervention included three visits, with the first visit consisting of face-to-face training using the teach-back method and the second and third visits to re-assess inhaler technique and reinforcement of face-to-face training if needed. After two educational sessions, the proportion of critical errors in inhaler technique dropped from 43.2 to 8.8% (*p* <  0.001). Results from this study by Ahn et al. are similar to effects seen on improvement in inhaler technique with implementation of this COPD care plan.

Results from this study add to existing literature by providing further insight into errors associated with administration technique with each type of inhaler including the pressurized metered dose inhaler (pMDI), dry powder inhalers (Diskus^®^ and Ellipta^®^ devices), the Respimat^®^ soft-mist inhaler, and the HandiHaler^®^. Implementation of this care plan was associated with higher rates of correct steps across all inhaler device types. In this study, patients had a higher baseline understanding of inhaler technique compared to previous studies [23]. This is attributed to the patients’ use of the inhalers prior to hospital admission.

The pMDI was the most common device type evaluated. Previous studies evaluating administration technique with the pMDI describe the most common errors with coordination of the dose actuation with inhalation and proper handling of the device [16,24]. In this study, the most common errors included lack of shaking the device prior to use; incorrectly repeating the steps if an additional dose was needed; and limited or lack of understanding of the purpose of the dose counter. The most common errors for the other device types display similarities to the pMDI. These errors are associated with poor understanding of how the devices worked, and suggest that patients may not have been provided with adequate inhaler education upon initiating these medications or a lack of reinforcement with continued medication refills.

### 4.2. Secondary Outcomes

During implementation of this COPD care plan, additional pharmacist interventions were noted. Of the 67 patients who received the COPD care plan, the pharmacist was able to make additional interventions for 25 patients. Based on patient chart review, evaluation of current inhaler regimen, and discussion with the patient, the pharmacist identified opportunities for medication optimization due to severity of disease and medication cost for 14 patients. In addition, the pharmacist identified patients in need of inhaler refills prior to discharge and made arrangements for refills where appropriate. The pharmacist was able to provide education regarding smoking cessation for six patients. A study conducted by van Boven et al. examined the effects of providing inhalation instruction, medication information, and motivational interviewing regarding medication adherence and smoking cessation to patients with suboptimal medication adherence and frequent COPD exacerbations. Although that study failed to demonstrate change in medication adherence, disease-related symptoms, or patient quality of life, it did demonstrate a significant reduction in COPD exacerbations [25]. Implementing this COPD care plan can allow for the opportunity to screen patients for further interventions, such as smoking cessation, medication adherence, immunizations, and need for additional services.

This study also highlights the pharmacist time spent implementing this COPD care plan. The majority of time was spent completing the checklist to obtain baseline assessment of inhaler administration and providing the initial inhaler education. Chart review and discussion of therapeutic interventions with the primary provider also encompassed a considerable amount of time. As described in a meta-analysis by Maricoto et al., there are limited studies which evaluate the time spent or cost-effectiveness associated with implementing an inhaler education program for patients with COPD [10,26]. A cross-sectional study conducted by Roggeri et al., sought to evaluate the potential economic impact related to inhalation errors after patients were switched to different inhaler devices without receiving proper education. This study concluded that patients who did not receive proper education had a higher cost of health care utilization through more hospitalizations, emergency room visits, and use of steroids and antimicrobials [17]. Additional studies are needed in order to compare the costs of implementing this COPD care plan with the health care utilization costs of patients with poor inhaler technique.

### 4.3. Strengths

There are several strengths to this study. The primary investigator conducted all pre- and post-assessments in order to avoid variability between investigators. Additionally, this study evaluated technique for all device types. Although there are limitations with subjective observation of inhalation technique, this represents a real-life setting, which supports this study’s external validity. A power analysis determined a sample size of 35 patients was necessary to identify a 25% change in the proportion of patients correctly identifying when to use short-acting versus long-acting inhaler device types. Although this more conservative power analysis was utilized, researchers sought to include a larger sample size in order to identify the differences between inhaler types. As such, this study highlights common administration issues with each inhaler type, and is consistent with previous studies [16,24,27]. This may be helpful when determining future medication therapies for patients. This care plan was provided to patients with an established diagnosis of COPD and prior use of inhalers. To our knowledge, this is the first study to evaluate the impact of a COPD initiative only with patients who have established disease and inhaler medications prior to hospital admission. This study demonstrates a continued need for education of inhaler administration and sheds light on the time spent implementing this care plan. In addition, this care plan could be implemented throughout other health-systems, and could be expanded and adjusted based on available staff and needs of the institution.

### 4.4. Limitations

Limitations include that participants in this study were not matched to an external control population. Therefore, it is unclear if other patient factors, such as comorbidities, could have influenced the impact of the COPD care plan. Inhalation technique was assessed using device type-specific checklists that were adapted from package inserts and other inhaler checklists [17]. These checklists were not validated, and may display some variability when evaluating use for specific inhalers. In addition, inhaler technique was subjectively observed by the primary investigator, which can lead to observer bias. Patients included in this study had a higher than expected baseline score on the checklist, with a median baseline score of 81.8% (67.5–97.0). This high baseline score makes it difficult to assess the true impact of this care plan on improving inhalation technique through using a scored checklist, as the follow-up score increased to a median of 100% (90.9–100). This study lacks evidence for effect on clinically significant endpoints, such as hospital readmission and exacerbation rates. Ongoing hospital initiatives including discharge planning services, medication delivery to bedside, and other independent research projects could have impacted these clinically significant outcomes; therefore, researchers sought to identify a primary endpoint that would not be confounded by other ongoing hospital initiatives. Although not evaluated in this project, this care plan could be implemented to assess its impact on multiple outcomes, including 30 and 90 day hospital readmission rates, morbidity, and mortality for patients admitted with COPD exacerbation.

### 4.5. Future Directions

This study describes a realistic pharmacist-led care plan that can be implemented within health systems. Along with improvement in patient inhaler technique, this study also provides insight into the pharmacist’s ability to help optimize management of COPD and the time needed to incorporate this service into daily responsibilities. Further research should evaluate the impact of this COPD care plan on 30 and 90 day hospital readmission rates.

## 5. Conclusions

Implementation of this pharmacist-led COPD care plan for patients admitted for COPD exacerbation was associated with an increase in correct steps associated with inhaler technique after pharmacist education and an increase in the proportion of patients able to identify when to use short-acting versus long-acting inhaler device types. This care plan also allowed additional interventions to improve care during a time period less than 1 h per patient. This study adds to the literature supporting interventions to improve patient inhaler technique and demonstrates a unique opportunity for pharmacists to address COPD-related transitions of care issues.

## Figures and Tables

**Figure 1 pharmacy-09-00202-f001:**
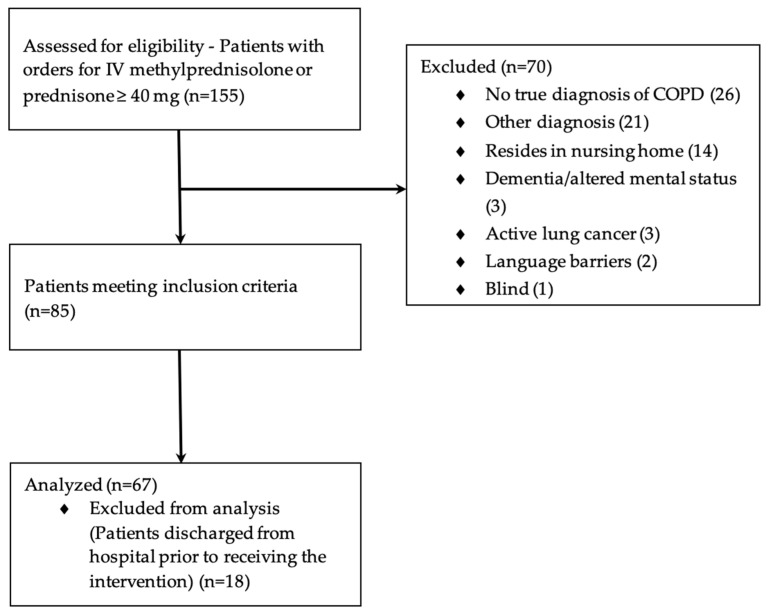
Participant Flow.

**Figure 2 pharmacy-09-00202-f002:**
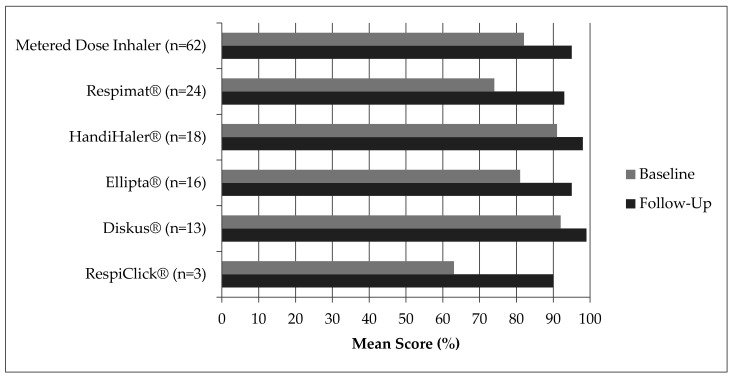
Mean Score (%) on Checklist Based on Inhaler Type.

**Table 1 pharmacy-09-00202-t001:** Intervention and Study Design Description.

Component	Intervention
Target population	Patients with a diagnosis of acute COPD exacerbation residing on the general hospital floors prior to discharge Patients admitted to the ICU were screened once transferred to the general hospital floors Patients with prior use of the inhaler device type prior to hospital admission
Patient identification	Patients with active orders for intravenous methylprednisolone or prednisone 40 mg or more with an indication of COPD exacerbation identified by institution report
Assessment of appropriate medication regimen	Based on GOLD Guideline criteria per the patient’s GOLD status Identify medications for optimization Inform medical team prior to patient discharge for correction
Individualized inhaler education and discharge counseling	Assess patient understanding of inhaler administration using scored checklist for each inhaler to determine baseline education Conduct face-to-face training session: Educate patient on proper inhaler technique, timing of inhalers, and proper storage Provide educational material to aid in inhaler education Reassess patient understanding of inhaler administration using scored checklist for each inhaler after education session Medication education will be documented in patient chart

**Table 2 pharmacy-09-00202-t002:** Patient Demographics and Clinical Characteristics.

Demographic	n = 67
Age (yr), mean (SD)	68 (±11)
Female, n (%)	30 (45)
Oxygen requirements prior to admission (L), mean (SD)	1.1 (±1.4)
Presence of comorbidities, n (%)	
Heart failure	21 (31)
Diabetes	13 (19)
Coronary artery disease	20 (30)
Hypertension	44 (66)
Hyperlipidemia	26 (39)
Obesity	19 (28)
Obstructive sleep apnea	19 (28)
Smoking history, n (%)	
Current	27 (40)
Previous	35 (52)
Never	5 (8)
COPD exacerbations in previous year, n (%)	
0	22 (33)
1	17 (26)
2	14 (21)
3	14 (20)
Inhaler medications prior to admission, n (%)	
LABA or LAMA	4 (6)
LABA and LAMA	7 (10)
LABA and ICS	14 (21)
LABA, LAMA, and ICS	42 (63)

Abbreviations: LABA, long-acting beta_2_-agonist; LAMA, long-acting muscarinic antagonist; ICS, inhaled corticosteroid.

**Table 3 pharmacy-09-00202-t003:** Patient Pre- and Post-Education Checklist Scores.

	Baseline	Follow-Up	*p*-Value
Median percent score on checklist ^a^ (IQR)	81.8 (67.5–97.0)	100.0 (90.0–100.0)	*p* < 0.0001
Proportion of patients correctly identifying when to use short-acting versus long-acting inhalers types ^b^, n (%)	49 (73.1)	66 (98.5)	*p* < 0.0001

^a^ Analyzed using Wilcoxon Signed-Rank test. ^b^ Analyzed using McNemar’s test.

**Table 4 pharmacy-09-00202-t004:** Inhaler Device Type-Specific Errors.

Most Common Incorrect Steps	Prior to Education n (%)	After Education n (%)	*p*-Value ^a^
Metered Dose Inhaler (pMDI) (n = 62)			
Hold inhaler upright and shake well	34 (55)	12 (19)	*p* < 0.0001
Repeat steps for additional dose if needed	33 (53)	12 (19)	*p* < 0.0001
Check dose counter	23 (37)	4 (6)	*p* < 0.0001
Respimat^®^ (n = 24)			
Repeat steps to get the full dose of two inhalations	15 (63)	8 (33)	*p* = 0.016
Hold inhaler upright with the cap closed	15 (63)	2 (8)	*p* < 0.0001
Close lips around mouthpiece; do not cover air vents	14 (58)	7 (29)	*p* = 0.016
HandiHaler^®^ (n = 18)			
Repeat steps to take the full dose	9 (50)	2 (11)	*p* = 0.016
Press green piercing button in once and release	5 (28)	0 (0)	*p* = 0.063
Close mouthpiece until it clicks	3 (17)	0 (0)	
Ellipta^®^ (n = 16)			
Check dose counter	7 (44)	4 (25)	*p* = 0.250
Slide cover down until it clicks; do not shake	6 (38)	2 (13)	*p* = 0.219
Close lips around mouthpiece; do not cover air vents	6 (38)	1 (6)	*p* = 0.063
Diskus^®^ (n = 13)			
Check dose counter	3 (23)	1 (8)	
Repeat inhalation to ensure dose is complete	3 (23)	0 (0)	
Hold breath for 5-10 s or as long as comfortable	2 (15)	0 (0)	
Respiclick^®^ (n = 3)			
Remove cap all the way down until you hear the click	2 (66)	0 (0)	
Breathe in slowly and deeply through mouth	2 (66)	0 (0)	
Repeat steps for additional dose if needed	2 (66)	2 (66)	

^a^ Analyzed using McNemar’s test when comparing incorrect steps before and after pharmacist education; not calculated for Diskus^®^ and Respiclick^®^ as sample size too small for a meaningful test.

**Table 5 pharmacy-09-00202-t005:** Assessment of Appropriate Therapy and Pharmacist Interventions.

Item	n = 67
Number of patients with pharmacist interventions, n (%)	25 (36)
Adjustment to medication regimen	12 (18)
Refills prior to discharge	11 (16)
Recommendation due to medication cost	2 (3)
Additional topics reviewed during education session, n (%)	
Smoking cessation education	6 (9)

**Table 6 pharmacy-09-00202-t006:** Pharmacist Time Spent Implementing the COPD Care Plan.

Item	Time (min)
Patient chart review and interventions prior to education, mean (SD)	15.9 (±2.8)
Baseline checklist and inhaler education, mean (SD)	16.6 (±3.7)
Follow-up checklist and reinforcement of inhaler education, mean (SD)	11.8 (±2.7)
Total time spent, mean (SD)	44.3 (±6.4)

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
