# Peer review of "Interventional, Quasi-Experimental Study of a Chronic Obstructive Pulmonary Disease Education Care Plan for Hospital Discharge"

_pharmacy, 2021, doi:10.3390/pharmacy9040202_

Round 1

Reviewer 1 Report

The research setting should be stated in the abstarct.

The care plan sounds to belong to 'nursing discipline'. I have not seen in any book or article in which the role of pharmacist is recognised as to develop the care plan or deliver care. You need to find a special word for it to prevent misunderstanding among readers. It is suggested to replace it with 'education plan or pharmacist education plan' throughout the text.

Methods section should be started with the recognition of the research design. You should use the equator to find out how an experimental study should be reported. CONSORT should be used as the checklist: Experimental studies | Study Designs | The EQUATOR Network (equator-network.org)

Table 1 reflects an educational plan rather than a care plan. As I stated, it should be rectified in the text.

The methods section is very disorganised. You should reorginise it through the use subheadings as follows: design, sample and setting, data collection, data analysis, ethical considerations. Bring related materials under each subheading.

The process of the development of the educational plan and the checkjlist and reliability and validity checking processes should be fully described.

The educational process and how it was implemeneted should be described practically to make it possible to repeat it. How have done what and how?

Recruitment and group assignments should be described with all details.

The CONSORT flow diagram should be drawn and added.

The discussion section should be improved as you have not compared your findings with those of other studies. Compare them point by point with those of previous studies in other settings.  A few citations have been used!

At the end of the discussion, limitations should be mentioned clearly.

Author Response

Thank you for your thoughtful review, please see the attached document with point-by-point response to your comments.

Reviewer 2 Report

This is an interesting and an important study on pharmacists’ interventions in COPD patients. I do not have any comments on the study methodology and the results. However, I have few suggestions for the authors in order to improve their manuscript:

  1. You should add more references to your manuscript and compare your results with previous studies. This topic is very important for clinical pharmacists and your manuscript should provide more data on previous results and measurements that could be implemented in clinical practice, or the measurements that have been acknowledged as successful. I have searched for the literature and found several studies, but I am sure there are plenty more:

Petite SE, Hess MW, Wachtel H. The Role of the Pharmacist in Inhaler Selection and Education in Chronic Obstructive Pulmonary Disease. J Pharm Technol. 2021;37(2):95-106.

Valentino AS, Eddy E, Woods Z, Wilken L. Pharmacist Provided Spirometry Services: A Scoping Review. Integr Pharm Res Pract. 2021;10:93-111.

Shiwaku E, Dote S, Kaneko S, Hei C, Aikawa M, Sakai Y, Kawai T, Iwatsubo S, Hashimoto M, Tsuneishi T, Nishimura T, Iwata T, Hira D, Terada T, Nishimura T, Kobayashi Y. Pharmacist involvement in the inhaler choice improves lung function in patients with COPD: a prospective single-arm study. J Pharm Health Care Sci. 2021;7(1):28.

Idowu O, Makhinova T, Quintanilha M, Yuksel N, Schindel TJ, Tsuyuki RT. Experience of Patients with COPD of Pharmacists' Provided Care: A Qualitative Study. Pharmacy (Basel). 2021;9(3):119.

Prosser TR, Bollmeier SG. Consistency of chronic obstructive pulmonary disease regimens for patients visiting community pharmacies with the Global Initiative for Chronic Obstructive Lung Disease recommendations. J Am Pharm Assoc (2003). 2021;61(3):299-307.

Li LC, Han YY, Zhang ZH, Zhou WC, Fang HM, Qu J, Kan LD. Chronic Obstructive Pulmonary Disease Treatment and Pharmacist-Led Medication Management. Drug Des Devel Ther. 2021;15:111-124.

Hudd TR. Emerging role of pharmacists in managing patients with chronic obstructive pulmonary disease. Am J Health Syst Pharm. 2020;77(19):1625-1630.

  1. add the study design in the title
  2. Line 11 – This is not background, and the sentence is not clear.
  3. Add the study setting in the abstract.

Author Response

(The authors gave the same response as above.)

Reviewer 3 Report

Title

  • “Implementation of an inhaler education COPD care plan for hospital discharge”; Please do not abbreviations in title.

Abstract

  • “Background: Assess the effects of a pharmacist-led COPD care plan that provided inhaler

education for administration prior to hospital discharge.” Please not start background with study aims.

  1. Introduction
  • Please cite more 5 to 10 references in introduction. These references must be cited and explained in discussion.
  • Please cite some systematic reviews and meta-analysis.
  • Please briefly explain “Chronic obstructive pulmonary disease (COPD)” in the first paragraph.
  • Second paragraph: “In response to increased hospital readmissions, the Centers for Medicare and Medicaid Services (CMS)”: How? Where? USA? and When?
  • Line 40: “bundled payments”? Please give more details.
  • WHO and FIP position?
  • Lines 58-64: please present this information at the end of discussion. For instance, create a section on study strengths.
  • “A pharmacist-led care plan focused on providing accurate inhaler education for administration was implemented at our institution.” What institution?

  1. Materials and Methods

- Please provide more details in methods.

- Materials and Methods are too compact. Please create more subheadings (e.g., participants, inclusion and exclusion criteria, prior to hospital discharge, after to hospital discharge, questionnaires/checklists, methods for primary/secondary endpoints, ethical approval, stastistical methodologies, etc.). Please consider the creation of a flow chart.

- How were the clinical data collected?

- Were data double checked? Quality control protocol?

- Pilot study?

- “COPD Care Plan for Patients Admitted with an Acute COPD Exacerbation”; Who developed this care plan? References?

- “the patient’s severity of COPD was classified using the 2019 Global Initiative for Chronic Obstructive Lung Disease (GOLD) Guideline recommendations”; Please give more details.

- Lines 111-112: “Scored checklists were prepared for each device type”; please present all checklists and developed materials in supplementary electronic files.

- “Face-to-face training on inhaler administration and educational handouts for each inhaler device were then provided by the pharmacist.” How? Duration? Please give more details.

- Please describe the type of studied inhalers in methods. Please cite all package inserts of the studied inhalers.

- The qualitative content of Tables 5 and 6 should be described in methods.

- Last paragraph of methods: references are missing; please present studies applying similar statistical methodologies.

  1. Results

- Materials and Methods are too compact. Please create more subheadings. The created subheading may be similar for methods/results and discussion.

- Table 4. Inhaler Device Type Specific Errors. Please describe the type of studied inhalers in methods. Please cite all package inserts of the studied inhalers. Please present a column with the statistical of the test. Please check APA guidelines for the presentation of stastistical data. Why is the name of the inhaler device underlined?

  1. Discussion

- Discussion is too compact. Please create more subheadings. The created subheading may be similar for methods/results and discussion. This structure will ensure the discussion of all topics. Have you discussed all topics?

- At least the following 3 sections should be created: effects of a pharmacist-led COPD care plan that provided inhaler education for administration prior to hospital discharge and primary and co-primary endpoint.

- Please cite more references in discussion. For instance, the new cited references in introduction.

- Please present a section on practical implications, future research, and study limitations (e.g., possible study biases) at the end of discussion. For instance, see https://www.ncbi.nlm.nih.gov/pmc/articles/PMC2917255/

  1. Conclusion

- Conclusion should reply to study aims and primary and co-primary endpoints. Please do not provide results in conclusion. Other information should be placed in discussion.

References

  • Please check the format of all references in instructions for authors.

Author Response

(The authors gave the same response as above.)

Round 2

Reviewer 1 Report

1. What you present as a document for calling your intervention as 'care plan', is only a statement that is not peer-reviewed. We should prevent confusing the article readers. You could call it, the pharmacist-led COPD management plan etc. By the way, if you insist, under the design or intervention, you should add your own definition of 'care plan'. You need to add: In this study, the pharmacist-led COPD care plan consisted of the following aspects......

Do not forget to cite to that statement, also.

2. Which type of quasi-experimental research is this? Clarify it under the design section.

3. All details with regard to a quasi-experimental studies' reporting should be clearly stated. For instance, the practical recruitment process and the data collection require elaboration. Use the following checklist and prepare the article accordingly:

https://jbi.global/sites/default/files/2020-07/Checklist_for_Quasi-Experimental_Appraisal_Tool.pdf

Author Response

Please see attachment with response to reviewer.

Reviewer 3 Report

In general, the authors have carried out all suggestions.

The quality of the paper has been improved. Congratulations.

Minor comments:

Abstract

  • “investigator recorded initial inhaler technique”; audio or video? Both?

2.5 Data Collection

Confidentiality of data? How long data will be stored?

Methods

Please include a section/Table/Annex with the characteristics/figures of all inhalers (see Figure 2). Please note that the present paper cover an international audience and these inhalers (see Figure 2) are only market in some countries.

Figure 1. Please give more details about “no true diagnostic”. Why? 26 patients? Why did 26 patients not have a correct diagnostic? So many….

Tables

  • Please review the format of Tables in the instructions for authors and in published papers.
  • Usually, tables not present lateral lines.

P-value or p-value? Please check all the paper

Conclusion

Please update the conclusion, regarding point the Secondary Outcomes (see point 4.2).

Author Response

(The authors gave the same response as above.)

Round 3

Reviewer 1 Report

Nothing more.